A review of raptor carcass persistence trials and the practical implications for fatality estimation at wind farms

Wilson David 1 david.wilson@thebiodiversityconsultancy.com
Hulka Simon 2
Bennun Leon 1 3
1 The Biodiversity Consultancy , Cambridge , United Kingdom
2 Abbotsbury , United Kingdom
3 Conservation Sciences Group, Department of Zoology, University of Cambridge , Cambridge , United Kingdom
Anderson Todd
Electronic publication date: 2022 Nov 15
Publication date: 2022
Volume: 10
Electronic Location ID: e14163
Received 2022 Apr 1; Accepted 2022 Sep 12
Copyright: © 2022 Wilson et al.
Copyright year: 2022
Copyright holder: Wilson et al.
License: This is an open access article distributed under the terms of the Creative Commons Attribution License, which permits unrestricted use, distribution, reproduction and adaptation in any medium and for any purpose provided that it is properly attributed. For attribution, the original author(s), title, publication source (PeerJ) and either DOI or URL of the article must be cited.
License URL: https://creativecommons.org/licenses/by/4.0/

Keywords: Wind energy, Carcass persistence, Raptor, Fatality estimation, GenEst, Vulture

Funding: The authors received no funding for this work.

==============================
Bird and bat turbine collision fatalities are a principal biodiversity impact at wind energy facilities. Raptors are a group at particular risk and often the focus of post-construction fatality monitoring programs. To estimate fatalities from detected carcasses requires correction for biases, including for carcasses that are removed or decompose before the following search. This is addressed through persistence trials, where carcasses are monitored until no longer detectable or the trial ends. Sourcing sufficient raptor carcasses for trials is challenging and surrogates that are typically used often have shorter persistence times than raptors. We collated information from raptor carcass persistence trials to evaluate consistencies between trials and assess the implications of using persistence values from other studies in wind facility fatality estimates. We compiled individual raptor carcass persistence times from published sources along with information on methods and location, estimated carcass persistence using GenEst and ran full fatality estimates using the carcass persistence estimates and mock datasets for other information. We compiled results from 22 trials from 17 sites across four terrestrial biomes, with trials lasting between 7 and 365 days and involving between 11 and 115 carcasses. Median carcass persistence was estimated at 420 days (90% confidence interval (CI) of 290 to 607 days) for the full dataset. Persistence time varied significantly between trials (trial-specific persistence estimates of 14 (5–42) days to 1,586 (816–3,084) days) but not between terrestrial biomes. We also found no significant relationship between either the number of carcasses in the trial or trial duration and estimated carcass persistence. Using a mock dataset with 12 observed fatalities, we estimated annual fatalities of 25 (16–33) or 26 (17–36) individuals using a 14- or 28-day search interval respectively using global dataset. When using trial-specific carcass persistence estimates and the same mock dataset, estimated annual fatalities ranged from 22 (14–30) to 37 (21–63) individuals for a 14-day search interval, and from 22 (15–31) to 47 (26–84) individuals for a 28-day search interval. The different raptor carcass persistence rates between trials translated to small effects on fatality estimates when using recommended search frequencies, since persistence rates were generally much longer than the search interval. When threatened raptor species, or raptors of particular concern to stakeholders are present, and no site-specific carcass persistence estimates are available, projects should use the lowest median carcass persistence estimate from this study to provide precautionary estimates of fatalities. At sites without threatened species, or where the risk of collision to raptors is low, the global median carcass persistence estimate from this review could be used to provide a plausible estimate for annual raptor fatalities.

Introduction

Wind energy is predicted to supply more than a third of global energy generation needs by 2050 (International Renewable Energy Agency (IRENA), 2019). While wind energy is considered ‘green’, it still has environmental impacts, the most obvious being collisions between birds and bats and turbines (Zimmerling et al., 2013; Ralston-Paton et al., 2018; AWWI, 2019). To provide an unbiased estimate of the number of fatalities occurring at a wind energy project, systematic fatality searches are conducted around a sample of turbines and the results adjusted to account for fatalities that occurred but were not found. One principal reason why fatalities are not found during searches are because they are removed by scavengers or decompose between searches (see Huso, Dalthorp & Korner-Nievergelt, 2017). To account for this type of bias, ‘carcass persistence’ experiments are conducted to estimate the probability that a fatality will remain observable at the next search (Huso & Dalthorp, 2014).

The standard approach for such experiments is to place sample carcasses at locations representative of habitats found in the fatality fall-zone around turbines, and monitor each to determine how long it persists (e.g., USFWS, 2012; Huso, Dalthorp & Korner-Nievergelt, 2017). Accurate estimates of carcass persistence are best achieved using fresh fatalities or carcasses of the same species or species group (e.g., raptors). Where this is not possible, carcasses of surrogate species are frequently used (e.g., Villegas-Patraca et al., 2012; Linnell & Smucker, 2019).

Birds of prey (hereafter raptors) are a group at particular risk from wind turbine collisions (e.g., Thaxter et al., 2017) and are often of high stakeholder concern. The use of non-raptor surrogates (e.g., domestic chickens) is not recommended for persistence trials as this underestimates persistence (e.g., Urquhart, Hulka & Duffy, 2015; DeVault et al., 2017). Calculating realistic fatality rates for raptors therefore requires carcass persistence estimates from trials that use raptor carcasses. Conducting such trials may be challenging at many sites. As predominantly apex predators, raptors are typically scarce and fatalities difficult to source in numbers required for persistence trials. Additionally, many of the large raptor species that are a principal concern at wind energy projects (e.g., vultures or eagles) have high conservation status and therefore can only be acquired from a limited range of sources (e.g., collision fatalitiies with wind turbines, vehicles or electrocution). Obtaining an adequate sample from such sources may take a long time and face other practical or legislative constraints. In this situation, options may be to wait until sufficient carcasses have accrued from project-related fatalities, organize collective trials involving adjacent wind farms, or use results from trials conducted elsewhere.

Here we evaluate raptor carcass persistence datasets from public studies worldwide to determine if there is consistency in raptor carcass persistence across studies, assess the implications of using persistence values from other studies in fatality rate analyses when field trials are not feasible, and to inform the design of fatality searches and future carcass persistence trials.

Methods

Data compilation

We used Google Scholar (searching for “carcass persistence” or “raptor persistence” with or without clarifiers–e.g., “wind”, “trial”, “removal”) to search for raptor carcass persistence records in published sources, including post-construction monitoring reports from operational wind farms. Cited documents within any published sources identified were also checked for relevance. Only studies (Table 1) where results were reported for each carcass were used in statistical analysis. If not reported directly, individual carcass persistence times were derived from graphs and document text when possible. For each carcass, we collated two key variables required for estimating persistence: the last time it was known to be present and the first time it was known to be absent. No attempt was made to adjust for each study’s potentially different interpretation of what defined a carcass as present or absent. For each site, we also determined the approximate location from report information and the terrestrial biome at the site using information from Dinerstein et al. (2017).

Table 1 Details of carcass trials compiled for this report.

Trial	Location on Fig. 1	Country	Biome	Carcasses	Trial duration (days)	Schedule (days)	Number of carcass checks	Median carcass persistence (days)	90% CI low (days)	90% CI high (days)	Reference	
Global		645		420	290	607		
Individual sites	
Altamont Pass 1992	1	USA	Temperate Grasslands, Savannas and Shrublands	30	7	2, 3, 7	3	18	8	41	(Orloff & Flannery, 1992)	
Altamont Pass 2005	17	180/360	‘at least every three days’	60	None scavenged	(Smallwood & Thelander, 2005)	
Altamont Pass 2010	11	291	Camera traps for 21 days, then ‘weekly’	59	14	5	42	(Smallwood et al., 2010)	
Big Horn WF	2	USA	Temperate Conifer Forests	28	63	1, 2, 3, 4, 7, 10, 14, 21, 28, 35, 42, 49, 56, 63	14	64	30	134	(Hallingstad et al., 2018)	
Bok Poort CSP	3	South Africa	Deserts and Xeric Shrublands	2	28	1, 2, 3, 4, 5, 6, 8, 10, 12, 14, 21, 28	12	Small sample size	(Jeal et al., 2019)	
Braes of Doune WF 2015	4	Scotland	Temperate Broadleaf and Mixed Forests	40	95	1, 3, 5, 10, 15, 20, 25,35, 45, 55, 65, 75, 85, 95	14	814	339	1,955	(Urquhart, Hulka & Duffy, 2015)	
Braes of Doune WF 2016	40	95	1,3, 5, 8, 10, 15, 20, 25, 30, 35, 40, 45, 55, 65, 75, 85, 95	17	551	243	1,249	(Urquhart & Whitfield, 2016)	
Campbell Hill WF	5	USA	Temperate Grasslands, Savannas and Shrublands	115	120	1, 2, 3, 4, 7, 10, 14, 20, 30, 40, 50, 60, 70, 80, 90, 100, 110, 120	18	246	162	375	(McClure et al., 2021)	
Diablo Winds WF	6	USA	Temperate Grasslands, Savannas and Shrublands	3	62	1, 2, 3, 6, 8, 10, 14, 20, 27, 34, 41, 48, 55 and 62	14	Small sample size	(Western EcoSystems Technology, Inc., 2006)	
Juniper Canyon WF	7	USA	Temperate Conifer Forests	12	63	1, 2, 3, 4, 7, 10, 14, 21, 28, 35, 42, 49, 56, 63	14	23	8	66	(Hallingstad et al., 2018)	
Montezuma II WF	8	USA	Deserts and Xeric Shrublands	27	28	1, 2, 3, 4, 7, 10, 14, 21, 28	9	75	33	173	(Hallingstad et al., 2018)	
Plum Brook Station	9	USA	Temperate Broadleaf and Mixed Forests	104	14	Camera traps used—continuous monitoring	14	31	20	48	(DeVault et al., 2017)	
Ponnequin WF	10	USA	Temperate Grasslands, Savannas and Shrublands	2	61	1, 2, 3, 4, 31, ~62	6	Small sample size	(Kerlinger, Curry & Ryder, 2000)	
Rangely oil field	11	USA	Deserts and Xeric Shrublands	29	38	1, 2, 3, 4, 5, 6, 7, then ‘every 3–5 days until they
disappeared or decayed beyond usability’	20	929	257	3,361	(Lehman et al., 2010)	
Spion Kop WF	12	USA	Temperate Grasslands, Savannas and Shrublands	27	365	1, 2, 3, 4, 7, 10, 14, 30	8	934	412	2,115	(Linnell & Smucker, 2019)	
Tomamae WF	13	Japan	Temperate Broadleaf and Mixed Forests	11	60	1, 2, 3, 4, 5, 6, 7, 8, 9, 10, 11, 12, 13, 14 and then every ‘two to four weeks’ until day 60	17	31	10	94	(Kitano et al., 2020)	
Top of the World WF	14	USA	Temperate Grasslands, Savannas and Shrublands	91	120	1, 2, 3, 4, 7, 10, 14, 20, 30, 40, 50, 60, 70, 80, 90, 100, 110, 120	18	1,586	816	3,084	(McClure et al., 2021)	
Trysglwyn WF	15	Wales	Temperate Broadleaf and Mixed Forests	11	≤60	Camera traps used—continuous monitoring	≤60	52	15	176	(Natural Power, 2021)	
Wolfe Island WF 2010	16	Canada	Temperate Broadleaf and Mixed Forests	14	14	either: 3.5, 7, 10.5, 14 or 7, 14	3	165	38	719	(Stantec 2011a, 2011b, 2012)	
Wolfe Island WF 2011	12	14	either: 3.5, 7, 10.5, 14 or 7, 14	3	156	33	729	
Wolfe Island WF 2012	4	14	either: 3.5, 7, 10.5, 14 or 7, 14	3	Small sample size	(Stantec, 2014)	
Yaloak South WF	17	Australia	Temperate Broadleaf and Mixed Forests	15	271	Daily for 2 months, then weekly to 4 months and then opportunistically	68	588	174	1,986	(Bennett, 2019)	
Note:

Sites in italics had low sample sizes and were not included in GenEst analysis. WF, Wind Farm; CSP, Concentrated Solar Power.

Data analysis

We calculated carcass persistence using the R (R Core Team, 2018) package GenEst (Dalthorp et al., 2020), and, in general, followed the processes outlined in the user guide (Simonis et al., 2018). GenEst is used to estimate mortality by correcting the number of observed fatalities to account for searcher efficiency, carcass persistence and the area searched, and is the industry standard for fatality estimation at wind farms (e.g., Weaver et al., 2020; McClure et al., 2021). At three sites, Altamont Pass, Braes of Doune, and Wolfe Island, data were available for trials in different years and years were analysed separately. Four trials with <10 raptor carcasses each, and one trial where no raptor carcasses were removed, were included in the global analysis but excluded from all trial-specific analyses.

We initially ran the full dataset in the GenEst carcass persistence module to provide a ‘global’ estimate of median carcass persistence. We then re-ran the analysis within GenEst with trial and biome as explanatory variables to test if there were consistent trial- or regional-scale effects on carcass persistence and, using the best-supported model, derived trial-specific median carcass persistence estimates. Exponential, log-logistic, lognormal, and Weibull distributions were fit, with the best modelled selected using AICc within the GenEst package. All estimates are reported with a 90% confidence interval (i.e. using the 5% and 95% estimate values).

We tested for relationships between trial-specific median carcass persistence estimates and confidence interval range and the number of carcasses and total length of trial (the period over which carcasses were monitored) using Generalised Linear Models with normal distribution and identity link function in Past3 (Hammer, Harper & Ryan, 2001).

To explore how using different median carcass persistence values could affect fatality estimates, we generated a plausible mock dataset for the other GenEst modules required to estimate fatalities (description of required data can be found in Simonis et al. (2018), and a description of the mock data files are in File S8). We ran GenEst to produce fatality estimates using the mock dataset with the global median carcass persistence value and with median carcass persistence values for individual trials, and applying either 14-day or 28-day search schedules to bracket the search frequency range recommended by the USFWS Wind Energy Guidelines (USFWS, 2012).

Results

We compiled results from 22 trials from 17 sites (11 from the USA and one each from Canada, Wales, Scotland, Japan, Australia and South Africa: Table 1, Fig. 1), and the full dataset is available as Supplemental Material (Table S1). Trials included in only the global analysis were from the USA (3), Canada (1) and South Africa (1) (Table 1). Sites were located in four terrestrial biomes: Temperate Grasslands, Savannas and Shrublands (6), Temperate Broadleaf and Mixed Forests (6), Temperate Conifer Forests (2) and Deserts and Xeric Shrublands (3). Trials included in the analysis lasted between 7 and 365 days and involved between 11 and 115 carcasses. Three trials used camera traps to monitor carcass persistence, while all other trials involved checking by observers, initially daily then decreasing to a frequency of every 3–14 days for the remainder of the trial period (Table 1).

Figure 1 Locations of sites included in this analysis.

Numbers correspond to sites listed in Table 1.

The full dataset estimated a global median carcass persistence time of 420 days (90% confidence interval of 290 to 607 days, Table 1; lognormal best-fit model, l ~ constant, s ~ constant, see Table 2 for full model outputs). When explanatory variables were added to the model, carcass persistence time varied significantly between trials but not between biomes (lognormal best-fit model, l ~ trial, s ~ constant: full model outputs in Table 2). Estimated trial-specific median carcass persistence values varied from 14 (5–42) days to 1,586 (816–3,084) days, with predictions for carcass persistence times in all trials shown in Table 1. Across the 17 individually analysed trials there was no significant correlation between the number of carcasses in a trial and the median carcass persistence (G = 1.45, p = 0.23) or 90% CI spread (G = 0.06, p = 0.80), nor between the duration of monitoring of individual carcasses in a trial and the median carcass persistence (G = 2.05, p = 0.15) or 90% CI spread (G = 1.23, p = 0.27).

Table 2 GenEst carcass persistence model outputs for the full dataset, and with trial as the predictive variable.

Not all models were successfully fit with trial as a predictive variable.

Distribution	Location formula	Scale formula	AICc	deltaAICc	
Global model	
lognormal	l ~ constant	s ~ constant	1,732	0	
loglogistic	l ~ constant	s ~ constant	1,744.62	12.62	
Weibull	l ~ constant	s ~ constant	1,751.83	19.83	
exponential	l ~ constant	NULL	1,897.57	165.57	
Models with trial and biome as covariates	
lognormal	l ~ Trial	s ~ constant	1,600.28	0	
loglogistic	l ~ Trial	s ~ constant	1,604.71	4.43	
lognormal	l ~ Trial	s ~ Biome	1,605.37	5.09	
lognormal	l ~ Trial + Biome	s ~ constant	1,606.77	6.49	
loglogistic	l ~ Trial	s ~ Biome	1,610.47	10.19	
loglogistic	l ~ Trial + Biome	s ~ constant	1,611.21	10.93	
lognormal	l ~ Trial + Biome	s ~ Biome	1,611.92	11.64	
weibull	l ~ Trial	s ~ constant	1,613.83	13.55	
loglogistic	l ~ Trial + Biome	s ~ Biome	1,617.02	16.74	
weibull	l ~ Trial	s ~ Biome	1,619.04	18.76	
weibull	l ~ Trial + Biome	s ~ constant	1,620.33	20.05	
weibull	l ~ Trial + Biome	s ~ Biome	1,625.6	25.32	
exponential	l ~ Trial		1,633.74	33.46	
exponential	l ~ Trial + Biome		1,640.22	39.94	
weibull	l ~ constant	s ~ Trial	1,675.36	75.08	
lognormal	l ~ constant	s ~ Trial	1,675.81	75.53	
lognormal	l ~ Biome	s ~ Biome	1,712.28	112	
lognormal	l ~ Biome	s ~ constant	1,712.62	112.34	
loglogistic	l ~ Biome	s ~ constant	1,721.6	121.32	
loglogistic	l ~ Biome	s ~ Biome	1,723.17	122.89	
weibull	l ~ Biome	s ~ constant	1,728.19	127.91	
lognormal	l ~ constant	s ~ Biome	1,730.16	129.88	
weibull	l ~ Biome	s ~ Biome	1,730.51	130.23	
lognormal	l ~ constant	s ~ constant	1,732	131.72	
loglogistic	l ~ constant	s ~ Biome	1,741.75	141.47	
loglogistic	l ~ constant	s ~ constant	1,744.62	144.34	
weibull	l ~ constant	s ~ Biome	1,744.91	144.63	
weibull	l ~ constant	s ~ constant	1,751.83	151.55	
lognormal	l ~ Trial * Biome	s ~ constant	1,759.36	159.08	
loglogistic	l ~ Trial * Biome	s ~ constant	1,763.8	163.52	
lognormal	l ~ Trial * Biome	s ~ Biome	1,766.09	165.81	
loglogistic	l ~ Trial * Biome	s ~ Biome	1,771.19	170.91	
weibull	l ~ Trial * Biome	s ~ constant	1,772.92	172.64	
weibull	l ~ Trial * Biome	s ~ Biome	1,779.77	179.49	
exponential	l ~ Trial * Biome		1,792.29	192.01	
exponential	l ~ Biome		1,834.2	233.92	
exponential	l ~ constant		1,897.57	297.29	

Using the combined global dataset, the mock GenEst dataset with 12 detected raptor fatalities, and either a 14-day or 28-day search interval resulted in an annual fatality estimate of 25 (16–33) or 26 (17–36) individuals respectively. Using trial-specific carcass persistence values with the same mock GenEst dataset and either a 14-day or 28-day search interval resulted in an annual fatality estimate of between 22 (14–30) and 37 (21–63) individuals or between 22 (15–31) and 47 (26–84) individuals respectively (Table 3). Moving from a 14-day to 28-day search interval increased the annual fatality estimate by a mean of three individuals (range 0–10) for the trial-based annual fatality estimate (Table 3).

Table 3 Annual fatality estimates based the mock dataset and global or site-specific median carcass persistence times.

Trial	Using a 14-day search interval	Using a 28-day search interval	Increase in fatality estimates by moving the search frequency from 14 to 28 days	
Global	25 (16–33)	26 (17–35)	1	
Sites	
Altamont Pass 1992	35 (21–55)	43 (24–71)	8	
Altamont Pass 2005	N. estimate—no carcasses scavenged	
Altamont Pass 2010	37 (21–63)	47 (26–84)	10	
Big Horn WF	28 (17–40)	31 (18–46)	3	
Bok Poort CSP	N. estimate—small sample size	
Braes of Doune WF 2015	22 (14–30)	23 (15–32)	1	
Braes of Doune WF 2016	23 (15–31)	25 (15–33)	2	
Campbell Hill WF	23 (15–32)	25 (16–35)	2	
Diablo Winds WF	N. estimate—small sample size	
Juniper Canyon WF	34 (20–52)	39 (22–67)	5	
Montezuma II WF	26 (17–39)	30 (18–44)	4	
Plum Brook Station	31 (18–46)	37 (21–55)	6	
Ponnequin WF	N. estimate—small sample size	
Rangely oil field	23 (15–31)	24 (16–32)	1	
Spion Kop WF	22 (14–31)	23 (15–32)	1	
Tomamae WF	32 (19–47)	37 (21–64)	5	
Top of the World WF	22 (14–30)	22 (15–31)	<1	
Trysglwyn WF	28 (17–43)	32 (19–52)	4	
Wolfe Island WF 2010	25 (16–36)	26 (17–41)	1	
Wolfe Island WF 2011	25 (16–36)	27 (17–42)	2	
Wolfe Island WF 2012	N. estimate—small sample size	
Yaloak South WF	22 (15–31)	24 (16–32)	2	

Discussion

There were large differences in raptor carcass persistence rates between trials, however these large differences had small effects on fatality estimates, with an average increase of three individuals when applying search frequencies of 14 and 28 days, as persistence rates were, in general, much longer than the search intervals. While site-specific information on carcass persistence is always preferable to use in fatality estimation, using the global carcass persistence estimate is unlikely to have a large effect on project-specific fatality estimates.

Trial-specific median carcass persistence estimates in this review ranged from 14 to 1,586 days, a seven-fold difference, which is consistent with the eight-fold difference between the daily carcass persistence probability found between sites by Borner et al. (2017) using red-legged partridge (Alectoris rufa) and common pheasant (Phasianus colchicus) carcasses. Only two studies for which we used data reported raptor carcass persistence rates: the Braes of Doune trial in 2015 and the Yaloak South trial (Urquhart, Hulka & Duffy, 2015; Bennett, 2019), although neither used GenEst in their analysis. The Braes of Doune 2015 trial reported an average persistence of 64 days, however as 85% of carcasses were still present on day 90, the true mean carcass persistence is likely to be much longer. The Yaloak South trial estimated persistence at 395 days (95% CI of 148–1,052 days), which is in general agreement with the estimates in this study of 588 (174–1,986) days. The upper estimate of persistence, at almost 5 years, is likely to be an artifact of the modelling process, and no carcasses are expected to last this long under natural conditions, although at least one carcass was still present 527 days after being placed in the field (Linnell & Smucker, 2019). A large variability in carcass persistence times is unsurprising, as carcass persistence rates are known to be influenced by both abiotic (e.g., temperature, rainfall: Santos, Carvalho & Mira, 2011; Bispo et al., 2013) and biotic conditions (e.g., abundance of scavengers or alternate prey, habitat: Borner et al., 2017; Peers et al., 2020). Both biotic and abiotic factors are almost certain to have been different at each of the sites in this review, given their large geographic spread, and is likely to explain some of the differences in trial-specific carcass persistence estimates.

Locations with strong environmental seasonality may also show markedly different carcass persistence rates between seasons, although there has been varying support for seasonal effects on raptor carcass persistence at individual sites (e.g., mixed: Orloff & Flannery, 1992; supported: Smallwood et al., 2010; unsupported: Urquhart, Hulka & Duffy, 2015; DeVault et al., 2017). Additionally, separate sites have shown contradictory effects of season on raptor carcass persistence: in Japan, estimated carcass persistence was shorter in winter than summer (Kitano et al., 2020) while the opposite was true at the Altamont, USA, site (Smallwood et al., 2010). While a ‘season’ is a grouping of broadly similar environmental conditions (see, e.g., Bispo et al., 2013), environmental conditions vary not only seasonally but on much shorter time scales. The environmental conditions on the days immediately following death that affect decomposition rates or palatability to potential scavengers are likely to have a greater effect than any ‘average’ seasonal or yearly conditions, which has implications for design of persistence estimation trials.

The trials analyzed in this study are restricted to sites in seven countries (Fig. 1, Table 1), three-quarters of which are from North America, and they represent sites in only four of the world’s 14 terrestrial biomes (Dinerstein et al., 2017). While biome did not have a significant effect on carcass persistence rates in this study, it is possible that average carcass persistence may be higher or lower in other parts of the world for which there is no information. Two regions of particular interest for comparison would be the African savannah, with its highly specialist scavenger guilds, and tropical areas with high humidity and/or rainfall: in such habitats median carcass persistence rates lower than recorded in this study may well be possible. Interestingly, Borner et al. (2017) found no effect of habitat on daily carcass persistence rates, even for sites <250 m apart. While available data are extremely limited, it may be that any ecosystem-level effects are insignificant when compared with site-specific conditions. Potential differences in carcass persistence in unsampled biomes or ecosystems may be ecologically interesting but are unlikely to have practical implications for fatality estimation unless median persistence times are shorter than the recommended search frequency.

The three sites with trials in multiple years showed similar estimates of median carcass persistence between years (14 v 18 days at Altamont: Site 1, 551 v 814 at Braes of Doune: Site 4, 156 v 165 days at Wolfe Island: Site 16, Table 1). Such a result would be expected if scavenger numbers at a site are (a) not primarily limited by prey availability, as has been suggested for apex predators (e.g., Wallach et al., 2015), (b) relatively constant across years due to long life-spans or (c) both. While ecological theory and the results from these three sites suggest that carcass persistence trials at a site may not be needed over multiple years, more multi-year carcass persistence trials would be needed to confirm this is the case. The lack of correlation between trial duration and median carcass persistence or estimate spread is not surprising. While there is likely to be a minimum duration and number of carcasses below which estimates are unreliable: once this is exceeded there is no obvious reason why more carcasses or longer trials would affect carcass persistence estimates.

While there were large differences between trials in median raptor carcass persistence times, these differences had relatively little effect on fatality estimates when using a plausible mock dataset–both for a 14-day or 28-day search frequency. This result is driven by the extremely long estimated carcass persistence times for most trials, as nearly all carcasses for nearly all trials would persist until at least the next search for both a 14 and 28 day search frequency. Moving from a search interval of 14 to 28 days resulted in only relatively small increases in both the mean annual fatality estimate (average of three individuals) and estimate uncertainty (CI of 23 and 28 for 14 and 28 days respectively). Our results suggest that a 28-day search interval is adequate for most sites where the goal is to provide a broad understanding of the project’s impacts. For species where a project has a permitted impact threshold, such as wind farms with ‘take’ permits for Bald or Golden Eagle (USFWS, 2016) or no net loss/net gain commitments (IFC, 2012; USFWS, 2016), a shorter search interval is likely to increase the confidence in estimates and provide more robust median values against which to assess compliance with such requirements. As raptors are often of high conservation status, a shorter search interval would be the preferred precautionary approach especially in any areas where no trials have previously occurred, or where much shorter carcasses persistence times might be predicted.

Conclusions

When threatened raptor species, or raptors of particular concern to stakeholders are present, and no site-specific carcass persistence estimates are available, projects should use carcass persistence data from lowest median estimate from this study (14 days: Altamont in 2010) to provide precautionary estimates of fatalities under a ‘worst case’ scenario. At sites without threatened species, or where the risk of collision to raptors is low, the global median carcass persistence estimate from this review could be used in fatality estimation to provide a plausible estimate for annual raptor fatalities at a wind facility. As neither biome (this study) nor habitat or proximity to existing trials (Borner et al., 2017) were found to influence carcass persistence values, there is no evidence that using trial-specific carcass persistence values from another project, based on the habitat similarity or proximity of the two sites, would provide a more accurate fatality estimate than using the global carcass persistence estimate.

Fatality estimates using this approach must still be interpreted with caution, especially for projects in biomes where trials have not been run (e.g., African savannah, tropical areas) that may have very different environmental conditions and potentially shorter median carcass persistence times than estimated here. We encourage raptor carcass persistence trials in under-represented areas, and the publication or public posting of both analysed results and raw data from further trials in any location.

Recommendation for carcass persistence trials based on this review

The findings of this review help to inform good practice for undertaking carcass persistence trials in several respects:

Timing of trials and individual carcass placement. Carcasses should be placed randomly at a wind facility throughout the period when collisions could occur for the species of interest (e.g., full year for resident raptors, during migration, the breeding or overwintering seasons). This will ensure that the range of carcass persistence rates recorded are reflective of the range of true carcass persistence rates and make best use of a limited supply of raptor carcasses.

Number of carcasses. We found no evidence that carcass numbers were correlated with median carcass persistence values, nor that more carcasses provided more precise carcass persistence estimates. Once above a minimum number of carcasses (Huso, Dalthorp & Korner-Nievergelt, 2017 recommends a minimum of 15–25 carcasses for each combination of factors that may influence persistence), logistical and practical considerations (e.g., the availability of carcasses or ability of field teams to check carcasses repeatedly) are likely to be the main determinant of the total number of carcasses included.

Frequency of carcass checks. The trials reported here mainly used similar check frequencies, typically aligning with checks on days 1, 2, 3, 4, 7, 10, 14 and then at weekly intervals until the trial was complete. In general, the more frequently that carcasses are checked the fewer carcasses are needed to achieve a particular precision level for persistence estimates (Huso, Dalthorp & Korner-Nievergelt, 2017). This needs to be balanced with the increased effort required of any field team, and camera traps may be an effective alternative to increase the monitoring effort (e.g., Smallwood et al., 2010; DeVault et al., 2017).

Duration of monitoring carcasses. The duration for which individual carcasses were monitored did not affect carcass persistence estimates or their precision. A more useful consideration might be ‘after what length of monitoring do trial data become useful?’, which may be best answered by considering how GenEst fits models to the data. GenEst compares four carcass persistence models (exponential, Weibull, log-logistic and lognormal: see Table 2). Exponential models are typically the best-fit model at short trial durations, but are often inaccurate as they assume that scavenging rate does not depend on carcass age (which is unlikely to be true: Bispo et al., 2013), and will frequently underestimate the scavenging rate for fresh carcasses (e.g., Dalthorp, Huso & Dail, 2017). Thus, carcasses should be monitored until the exponential model is no longer the best-fit model for the trial–this duration is likely to be different for each trial, and data should be analyzed regularly to determine when this occurs. As we also recommend that carcasses are placed throughout the period when collisions are expected, some carcasses are likely to be monitored for far longer than others before the exponential model is not the best-fit for the data.

Supplemental Information

Supplemental Information 1 Raw persistence times for individual raptor carcasses.

Click here for additional data file.

Supplemental Information 2 Searcher schedule for a 28 day search interval.

Click here for additional data file.

Supplemental Information 3 Search schedule for a 14 day search interval.

Click here for additional data file.

Supplemental Information 4 Searcher efficiency mock trial data.

Click here for additional data file.

Supplemental Information 5 Density-weighted proportion value for turbines.

Click here for additional data file.

Supplemental Information 6 Carcass observations for a 28 day search schedule.

Click here for additional data file.

Supplemental Information 7 Carcass observations for the 14 day search schedule.

Click here for additional data file.

Supplemental Information 8 Description of the mock GenEst dataset for fatality estimation.

Click here for additional data file.

Additional Information and Declarations

Competing Interests

Author Contributions

Data Availability

The authors declare that they have no competing interests.

David Wilson and Leon Bennun are employees of The Biodiversity Consultancy

David Wilson conceived and designed the experiments, analyzed the data, prepared figures and/or tables, authored or reviewed drafts of the article, and approved the final draft.

Simon Hulka conceived and designed the experiments, authored or reviewed drafts of the article, and approved the final draft.

Leon Bennun conceived and designed the experiments, authored or reviewed drafts of the article, and approved the final draft.

The following information was supplied regarding data availability:

The full carcass persistence dataset and the mock datasets for remaining GenEst modules are available in the Supplemental Files.

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
