# Peer review of "A review of raptor carcass persistence trials and the practical implications for fatality estimation at wind farms"

_PeerJ, doi:10.7717/peerj.14163_

## Round 0.1 · original submission · Major Revisions

The reviews on your manuscript are mixed, but even those that are supported have concerns about the methodology and the conclusions drawn from a limited amount of data. I would like to give you the opportunity to address the reviewer concerns, because they and I believe that the work is important in the context of raptor conservation. All three reviewers made a thorough assessment of the manuscript; there comments are clear, so I won't repeat them here. Please use the comments and modify your manuscript accordingly.

·

Basic reporting

The writing of this article is of high quality, with good introductory sentences that establish importance and present the material in a concise fashion. Methods are presented the same; writing is concise and reasonable. Results writing is fine, and the discussion could use some elaboration (see specific comments about justifications and evaluating results in more detail).

Without thinking about the results that are presented, the article appears professional and reasonable. I looked into several of the citations, which were all informative and useful for the subject. The supplemental data provided was helpful in evaluating the results.

The premise of this article is important for the wind energy community. In order to estimate fatality data, we need carcass persistence information, and coming up with an industry standard to use in place of facility-specific studies could be useful. But, that information needs substantial justification, and I do not feel this manuscript as written as met that goal. And, I am not sure editing will solve the problem, as the ultimate problem is a lack of data in available literature. But, if edited to appropriately suggest caveats and discuss the significance of results more substantially, this could be a first step in calling for better studies to get us closer to the goal of better carcass persistence information and thus better estimates of raptor fatalities at wind farms.

Experimental design

The method used is a black-box estimator. It will take any data input and export numbers with an unknown methodology of creating those numbers. At least, that is my understanding from the GenEst user manual cited. This method probably works fine with high quality data, but the data input in this case had 72% missing information. This is a situation of "junk in = junk out", meaning the model will estimate some number no matter what is input, but there is so little quality information being input that the resulting numbers should be considered highly suspect. And no attempt was made to justify the results - no citations, no addressing the issue that most results were far beyond the scope of studies cited, no caveats that maybe we don't have enough information yet, etc. Instead, the authors downplay the results and suggest their median number of 444 days should be used by wind energy managers worldwide with no suggestion that further studies could yield a better estimate in the future.

If the authors had better quality data, or if they tried using just the data with complete information, this method might be reasonable and produce high quality results. But, as the methods are described in this manuscript, I believe GenEst was an innappropriate methodology, or the data used should have been used differently. Or, at a minimum, a much greater discussion should have been made about the results they were able to create using the data they had available.

The other method of producing mock simulation data and then estimating fatality numbers for individual wind farms, along with testing the difference in 14-day or 28-day search plans, was fine and appropriate for this study (though I would have liked to see more detail on how this was estimated in the methods). The issue I have with that part is the discussion points made and lack of recognizing the potential significance of the results (see other comments).

Validity of the findings

Upon investigation into the methods and results, I have significant issues with this manuscript. The authors present substantial numbers of wide variability and downplay them as if they matter not to the issue at hand, which is the ultimate goal of estimating raptor fatalities using incomplete information at wind farms. The final conclusions are problematic; 1) they recommend wind managers use their results that came from a biased methodology without justification that the median is biologically reasonable, and 2) they present the fatality estimation as if using different persistence numbers made negligible difference, when their results show clear and substantial differences. I cannot recommend this article move forward with publishing without reevaluation of the methods and results, significant justification that those results are reasonable, and a better evaluation of how those numbers translate to practical wind energy management. If our goal is raptor conservation, then this article presents data that could hinder that cause by giving wind energy managers an opportunity to underestimate the number of raptors their turbines might be killing every year.

Most important comments on methods, results, and discussion:

Your carcass persistence numbers are astronomically high. You take studies that lasted 7 to 365 days and jump to the conclusion that some carcasses persisted on those sites for nearly 3 years, with a high end of estimation at 9 years, and that is supposed to be a median estimation, which assumes some carcasses would persist much longer? That is quite a leap in assumptions.
Some persistence numbers make sense, like the camera trap study that likely had good, detailed data, and your median came out to be 14 days for a study that lasted 291 days long. Then one study only checked carcasses for 14 days total, and you estimate their carcasses persisted for 160+ days? One study lasted 120 days long and your high end of a median estimate is nearly 9 years? Don’t bones break down at some point? What is there to persist for 9 years? How can you present this data as reasonable estimates?

I read through the GenEst user manual, and their example data has only one carcass that persisted beyond the hypothetical study. The data you submitted in supplemental file S1 has only 177 carcasses with first-gone dates, out of 634, which means 72% of your data has infinite persistence data. I believe that is significantly biasing your results. Some specific examples – Bracs of Doune 2015 – 6 carcasses were removed by day 30, and 34 persisted beyond the study of 95 days; you estimate they persisted for 835 days, with a low estimate of 334 days? That is so far beyond the scope of the study. Rangely oil field – study lasted 38 days and you estimated 959 days persistence? That data is labeled as “Colorado” in Table S1, which made it difficult to find, but there are 3 carcasses with data and 26 inf. I understand that you are loading what data you have available into an R Package, which is apparently a black box estimator. But you don’t find these results a bit suspicious? It is bizarre to draw conclusions that might have occurred years after a study stopped monitoring the carcasses.

Did any of these studies estimate carcass persistence days? If so, does your data line up with theirs? Is there some argument to support that these numbers are reasonable? Did you try estimating persistence using only complete data to compare? Is there any evidence in decomposition literature that bird carcasses would persist at a median of 444 days?

And more importantly, is it appropriate to recommend the wind energy community use a general estimate of 444 days, or any of these table of numbers, when they go to make their fatality estimations, if they do not have carcass persistence trials conducted of their own? I argue the data is so biased towards a lack of information that the GenEst program is just wildly guessing at carcass persistence numbers and needs far better data input before it can get anywhere close to a real estimate. And, therefore, it is nonsense to conduct a generalized “global” estimate using 72% unknown data and produce a number that you recommend to others to use in their calculations that impact raptor conservation, policy making, and accountability at wind energy facilities.

What I find interesting is how the different persistence estimates appear to affect the fatality estimates. Those with the lowest persistences had the highest estimates of fatalities. Which makes sense; if carcasses are disappearing and searchers are failing to detect them, then that should inflate the estimate. And when carcasses persisted beyond a year, fatality estimates settled around 22-23 birds, which means searchers would fail to detect just under half the carcasses available after a year of searching. Perhaps that is reasonable. But, the estimate also relied on searcher efficiency data and other mock data that was input. This tells us that perhaps carcass persistence makes no difference if a median really is beyond a year. But, I argue that we do not know a median persistence timeframe, because your data are based on 72% missing information, and you present no citations of other researchers that have attempted to estimate carcass persistence time.

If you are able to justify the carcass persistence results as being reasonable, then I ask that you 1) address that the results are often higher than the length of the studies (you mention it in lines 165-166, but it sounds like no big deal the way it is worded) and address your justifications for concluding those are reasonable estimates in the discussion. Citations would be very helpful.

And 2) address the effect of persistence numbers on fatality estimates. I had to make myself a table comparing the two to draw the conclusion there is a clear pattern myself. Make that clear for the reader, because it may be of some interest, especially for those regions where carcass persistence is very low due to high scavenging or decomposition rates. In line 167, you basically tell the reader using an estimate of 444 days would have little effect on your site-specific fatality estimate, which I argue is not true. The difference between estimates using persistence numbers >1 year and those using smaller numbers is 16 and 23 raptors killed in your mock study. That is a huge difference. If my study site has a carcass persistence of 14 days, but I don’t know that and I use your 444 days, I am missing upwards of 23 birds in my fatality estimates, assuming everything else is the same as your mock data.

Using an inappropriately large carcass persistence number without good justification and well-supported background data could greatly underestimate the number of raptors being killed, which could detriment conservation efforts worldwide. I therefore cannot recommend this article move forward with publishing until substantial effort is made to justify the results, or results are improved with better data.

Line 215 – 216 – You presented data that suggests carcasses persist in environments for well beyond a year, upwards of nearly 9 years at one site (and that is just a median, which implies some would persist much longer). And then you say that it is possible that carcass persistence trials don’t need to be conducted over multiple years. I argue that if studies are finding carcasses to still exist on the landscape beyond 365 days, they need to keep following those carcasses to see how long they actually persist. We need multi-year data to confirm if raptor carcasses can actually survive as long as you estimate, and what a median would actually be, along with record longevity. Only then would you have something to compare these data you present to. If studies continue to just stop when logistically reasonable, instead of carrying out a persistence trial until every carcass has a conclusion, then biased estimates that I comment on above will be the only thing the wind energy industry has to work with.

Lines 218 – 221 – You state that once some minimum number of days and number of carcasses is surpassed, it becomes basically irrelevant to have increasing durations or numbers of carcasses in trials, citing some resource that recommends a minimum number of carcasses. But your persistence estimates vary widely, from 14 to 1629 days. There were studies that you analyzed that had well beyond the suggested number of carcasses and upwards of a year of data collection and your own estimates are highly variable among them. I do not see how you can have the results you produced and state this statement. And, my argument above is that longer trials are absolutely needed to confirm if persistence numbers should be as high as you estimate them in this study, at least until we have better data on this subject. After more data is collected, perhaps some industry standard can be put into place, but I argue we are far from that at this time, at least based on what you’ve presented here.

Line 229 – you state that all cases had <3 bird difference between 14 and 28 day trial, but your Table 3 shows 7 cases where the difference was 4-8 birds. Did you mean ‘most’ cases? Shouldn’t there be some discussion of why some cases had such large differences, and if that matters for wind energy management or raptor monitoring?

Additional comments

More minor comments:

I do not understand how you got fatality estimates from carcass persistence trials. From reading your paper several times, I get that you got the number from 12 hypothetical detections plus the carcass persistence information. But, I do not get how the carcass persistence information corrects the bias in detections or produces a new, larger number of potential raptor kills. For someone who is not familiar with the details of wind energy statistics, it would be nice to be provided a bit of information on how carcass persistence trials translate to the more-important estimation of raptor fatalities.

Line 22 – “where carcasses are monitored are no longer detectable” – sounds like a typo. Perhaps you meant, “where carcasses are monitored until no longer detectable”.

Line 23 – I think “that” should be “than”

Line 24 – “using raptors carcasses” – I believe the word raptor is meant to be singular.

Line 103 – Shouldn’t Program R be mentioned along with a version number?

Line 106 – “for trails in different years at and we analysed” – I think the ‘at’ is a typo.

Line 106 – Why not use <10?

Line 164 – I would leave out the word “quite”.

Line 170 – You say median carcass persistence estimates ranged from 14 to 959, but your Table 1 has one estimate of 1629 days.

Line 173 – 174 – I argue the range is quite surprising. As a reader with little background in wind energy fatality statistics, I find the range of 14 days to 3-9 years being quite a surprising stretch of variability. And I am not sure why you play it down like these numbers are negligible. Please see my further comments on this data above.

Lines 194 – 208 – You mention the world’s 14 terrestrial biomes, but I am not sure all 14 could be relevant to wind energy research. How many of those biomes have wind energy farms? How many more of them could have wind energy in the future? You specifically mention African savannahs and tropical rainforests. How many wind farms are in those locations? I have no idea about Africa, but the rainforests of Central and South America are largely of poor wind resources and only contain wind farms at a few locations in mountainous and coastal regions. I don’t think the discussion paragraph is unwarranted, given that you need to address using research from different biomes. I just wanted to point out that perhaps not all biomes are relevant.

Table 1 – I’m not sure how PeerJ prefers tables formatted, but generally few outlines are used throughout tables. When I construct tables, its usually a horizontal line above and below the whole table, and one below the heading names, and usually that’s it unless an extra line is warranted somewhere. But, your table is so complex, these outlines may be appropriate.

Table 2 – Same comment, but additionally, some mis-formatting has occurred causing a section of the table to me mis-aligned with the rest.

Figure 1 – You might want to darken the grey a bit. When I printed the document, only the points showed up.


Upon re-reading my own comments I'd like to add a suggestion: Instead of this manuscript as written, the authors could take this in a different direction that would be more reasonable, and potentially more useful for the raptor conservation community. Write a manuscript evaluating the validity of using GenEst with incomplete carcass persistence data. Evaluate the range of numbers produced with varying study timeframes, carcass numbers, and number of carcasses that were not followed to conclusion. Call for more studies to be conducted that give us more complete information. Call for reforms of the carcass persistence trials, or that an industry standard could be produced if a few studies chose to continue monitoring until every carcass is gone. And, more importantly, explore the variability in fatality estimation when managers are working with such incomplete data. Rather than downplay the results you have here, emphasize the differences. Try more mock data with larger turbine fields - what happens when you have the same information but it is 200 turbines? What happens when it is the same number of turbines, but the number of carcasses found varies? Emphasize that site-specific information is important, because things like variable carcass persistence do impact the fatality estimates. This subject is important to explore, and taken in a different direction, could be a really interesting study.

Reviewer 2 ·

Basic reporting

- Overall, I found the manuscript to be well written and free of grammatical and other errors. The writing style employed is, in my opinion, economical and the manuscript is free of superfluous verbiage and sentences.
- The author’s introduction is a brief but thorough review of the topic and provides the necessary information to bring non-specialists or non-experts up to speed on the issue/problem. Furthermore, the introduction clearly outlines the problem and clearly justifies the need for this study. Literature references are of the correct number and time period.
- I found the figure depicting the locations of the study to be useful and, though it shows a bit of a bias against tropical areas, this isn’t the fault of the authors and the absence of studies in that areas highlights a gap in our knowledge.
- The manuscript’s tables are a bit large and there might have been a formatting error on table 2 (columns were not aligned). However, this may have been an issue only on my computer.

Experimental design

- I found the search methods employed by the authors to be objective and sound. Their exclusion of records not reporting statistics for individual carcasses appears reasonable given their objectives.
- I am curious to know how the authors handled variation in persistence due to decay. For example, different studies of carcass persistence could evaluate the same carcass as remaining or removed depending on how the experimenters classified decaying carcasses. At what point might a carcass be considered no longer present? Remains of bones? Feathers?
- I found the statistical treatment of data to be sound, although I believe that readers would benefit from a more complete description of what GenEst does statistically. As is written, an unfamiliar reader is left to infer this from the results.

Validity of the findings

- In my assessment, the findings of this manuscript hold up to scrutiny. Tables provide a clear view of models considered and their output.
- Further, the analysis of the mock data set with results from the top model show the applicability of the findings.
- Because the article is a mix of review and novel analysis, the quality and quantity of the data are out of the author's hands. Still, despite this the author's have done the best possible to ensure uphold the standards of this journal and of modern science.

Additional comments

Wilson et al. review studies on the persistence of raptor carcasses at wind energy facilities to evaluate consistencies between trials and highlight possible differences in persistence across biomes and other factors. Using data from previously published works, they arrive at a global median persistence rate that could be used to estimate raptor persistence where site-specific data are lacking. Such estimates are needed as wind energy facility construction continues as demand for renewable energy increases. I enjoyed reading your manuscript.
- Abstract
o Paragraph 1, line 22. The phrase “…where carcasses are monitored are no longer detectable.”, seems to be missing a word or two.
- Introduction
o Overall, a sound introduction that nicely brings the reader up to speed.
- Methods
o Data compilation. I am curious to know if there was information on at what point a decaying carcass was considered no longer present. I would imagine that reasonable persons could disagree about whether a carcass was still present based upon the materials (bone, feather) that remained. Was there any effort to attempt to control for this? I only ask because some of the persistence times were much longer than I would have expected (e.g., 1629 days from McClure et al.). If standards of what has and what has not persisted differed among manuscripts reviewed, this could mask important effects or skew results.
o Data analysis, paragraph 1 line 105. Add an Oxford comma following Braes of Doune.
o Data analysis, paragraph 1, line 106. Delete the word “at” following the phrase “trials in different years”.
o Data analysis, paragraph 2, line 111. As written, this line uses “trial” where as elsewhere “site” seems to be use for the location.
- Results
o Straightforward. No substantial comments.
- Discussion and conclusion
o No substantial comments. Discussion and conclusion seem reasonable based upon the data collected and the analyses performed.

Reviewer 3 ·

Basic reporting

A proofread should be conducted, to fix missing words, commas, and inconsistencies between text and tables/figures, including:
Lines 21-22: “This is addressed through persistence trials, where carcasses are monitored are no longer detectable”. Add “until they” after “monitored” or revise as needed.
Line 35: change “site specific” to “site-specific”
Line 60: “Huso et al. 2017” vs “Huso, Dalthrop & Korner-Nievergelt, 2017”. Make these consistent.
Line 69: add comma after “possible”
Line 88: add “across studies” after “in raptor carcass persistence”
Lines 93-95: According to this text, the search was conducted for “wind”; however, Table 1 shows results for solar and oil. Please explain. Also, “carcass removal” might be a search term of interest and could result in additional studies to increase the sample size.
Line 106: delete “at” in “for trials indifferent years at and we analysed…”
Line 106-107: “Four trials with less than 10 carcasses each” please state whether this was total carcasses or raptor carcasses only. Provide justification for excluding studies with less than 10 carcass trials. Also, please make sure that the use of “trial” in this sentence is accurate. According to Table 1, the studies (not the trials) were excluded from the site-specific analysis if they had less than 10 persistence trial carcasses. Please revise as needed.
Lines 106-109: According to Table 1, an additional study (Altamont Pass 2005) was excluded from the global analysis because no carcasses were scavenged. Therefore, of the 21 total studies (or trials, as used in this manuscript – see Line 130 – ) four were excluded due to less than 10 carcasses, and one due to no scavenging. Add this to text.
Line 108-109: “these trials involved raptor carcasses placed as part of broader trials rather than trials specific to raptor carcasses”. Were all the other studies specific to raptor carcasses? If only studies that provided results for each carcass (regardless of carcass type) were used, why is this relevant for this study?
Line 132-133: refer reader to Table 1 at the end of this sentence
Lines 135-136: According to table 1, 3 sites (not 2 as stated in text) were located in the Desert and Xeric Shrublands Biome.
Line 157: Provide units for fatality estimates. Number of individuals? Number of individuals/MW/year? Per turbine? Provide units here and in Table 3
Line 163-164: Please state clearly, how the results support the statement that “… these large differences had quite small effects on fatality estimates when using…”
Line 212: According to text “…Wolfe Island: Site 11”; however, according to Table 1, Wolfe Island is Site 15
Lines 228-229: Compare with Lines 158-159 where it was stated that “Moving from 14-day to 28-day search interval increased the annual fatality estimate by a mean of 3 individuals (range 0-8)”. Is it three individuals? Less than three individuals? In all cases? In addition, where is the “slight increase” in the “estimate uncertainty” shown in the results?
Line 268: add “and” after “… carcasses repeatedly)” or revise accordingly (something is missing in this sentence)
Table 1: Suggest adding study period (i.e., dates) to the information presented in this table. Also, suggest using “studies” instead of sites, because some sites (for example Altamont) had studies (for example the one conducted in 2005) that were not included in the site-specific analysis

Experimental design

Address comments above, as needed.
Research question was well defined, on a topic that is relevant to post-construction fatality studies

Validity of the findings

Address comments above, as needed.
Uncertain on how the results support some of the statements made. This section needs work

---

## Round 0.2 · accepted · Accept

Thank you for your efforts to address reviewer comments and revise your manuscript accordingly. I believe the GenEst model concerns have been adequately addressed. Your manuscript is now ready for publication.

·

Basic reporting

No comment

Experimental design

The authors have addressed several concerns previously expressed, and the manuscript is much improved.

I do feel several of my previous concerns still stand regarding the results of the GenEst modelling. I understand that 1) I do not know this modelling procedure and 2) it is meant to produce results with the unknown data. I made my statements based on my general understanding of models, which is the result is only as good as the input and that the more ‘missing’ or ‘unknown’ data that is put in, the more it will impact the results. The authors seem to dismiss those comments as being irrelevant, and I will acknowledge that they are working with the best available data (which is not enough given the poor reporting in literature) and the best available model to interpret that data. However, those are still concerns using this method, regardless of whether or not this method is the gold standard for current wind energy research. I remain skeptical of the numbers produced from the GenEst model, and hope other readers realize that there needs to be some level of skepticism regarding what is presented. And I raise it as a concern, because many will not – they will take the numbers at face value with no regard to understanding that these are modelled results and not necessarily realistic expectations of carcass persistence. That is just an understanding of human psychology – skimming articles and receiving little information with low critical thinking is a common behavior.

Validity of the findings

My concerns with validity are the same as those with the experimental design.

Additional comments

I appreciate that the sentence at lines 186-188 has been added. This provides some relief to my concerns – noting that artifacts of the modelling process should be considered when reviewing the numbers of the table. And noting the longest length of time a carcass has been known to still be in the field is important as well. I feel like the core of this sentence – that these results should be taken with a grain of salt – should be repeated in the abstract and the conclusion regarding the results of this GenEst study, because where it currently stands is only responding to a previous study that attempted to estimate persistence using different methods. This should be an important point regarding this current study.

However, changing the recommendation to use the 14-days estimate for specific circumstances where a conservative estimate is useful changes the impact of this article. This increases my confidence that the information may be used in a positive way for the wind energy industry.